# Leveraging natural language processing for efficient information extraction from breast cancer pathology reports: Single-institution study

Phillip Park[1,2], Yeonho Choi[2], Nayoung Han[3], Ye-Lin Park[2], Juyeon Hwang[2,4], Heejung Chae[2,5], Chong Woo Yoo[3], Kui Son Choi[2,6], Hyun-Jin Kim[2]*

**1** Department of Digital Health, Samsung Advanced Institute for Health Sciences and Technology, Sung Kyun Kwan University, Seoul, Korea, **2** Cancer Data Center, National Cancer Control Institute, National Cancer Center, Goyang, Korea, **3** Department of Pathology, National Cancer Center, Goyang, Korea, **4** Department of Health Informatics and Biostatistics, Graduate School of Public Health, Yonsei University, Seoul, Korea, **5** Department of Internal Medicine, Center for Breast Cancer, National Cancer Center, Goyang, Korea, **6** Graduate School of Cancer Science and Policy, National Cancer Center, Goyang, Korea

* hyunjin@ncc.re.kr

## Abstract

### Background

Pathology reports provide important information for accurate diagnosis of cancer and optimal treatment decision making. In particular, breast cancer has known to be the most common cancer in women worldwide.

### Objective

For the data extraction of breast cancer pathology reports in a single institute, we assessed the accuracy of methods between regular expression and natural language processing (NLP).

### Methods

A total of 1,215 breast cancer pathology reports were annotated for NLP model development. As NLP models, we considered three BERT models with specific vocabularies including BERT-basic, BioBERT, and ClinicalBERT. K-fold cross-validation was used to verify the performance of the BERT model. The results between the regular expression and the BERT model were compared using the named entity recognition (NER) techniques.

### Results

Among three BERT models, BioBERT was the most accurate parsing model (average performance = 0.99901) for breast cancer pathology when set to k = 5. BioBERT also had the lowest error rate for all items in the breast cancer pathology report compared to other BERT models (accuracy for all variables ≥ 0.9). Therefore, we finally selected BioBERT as

**Data availability statement:** The datasets generated and/or analyzed during the current study contain potentially sensitive information or risks of personally identifiable information. Access to the data can be obtained through reasonable request and review process by the Data Management Committee of National Cancer Center contact via email (data@ncc.re.kr). This access policy allows for data sharing for appropriate research purposes while considering the sensitivity of the data and adhering to ethical guidelines.

**Funding:** This study was supported by a grant from the National Cancer Center (grant no. 1810871-2 and 2310400-1). The funders had no role in study design, data collection and analysis, decision to publish, or preparation of the manuscript.

**Competing interests:** The authors have declared that no competing interests exist.

the NLP model. When comparing the results of BioBERT and regular expressions using NER, we identified that BioBERT was more accurate than regular expression method, especially for some items such as intraductal component (BioBERT: 1.0, RegEx: 0.1644), lymph node (BioBERT: 0.9886, RegEx: 0.4792), and lymphovascular invasion (BioBERT: 0.9918, RegEx: 0.3759).

## Conclusions

Our results showed that the NLP model, BioBERT, had higher accuracy than regular expression, suggesting the importance of BioBERT in the processing of breast cancer pathology reports.

## Introduction

According to the International Agency for Research on Cancer, breast cancer has become the most frequently diagnosed cancer globally, overtaking its second place ranking in 2018. Breast cancer is the fifth most common cancer in Korea, with a total of 23,647 cases, and was the most common malignancy among women (20.5% of all cancer cases in women) in 2018, according to the Korea Central Cancer Registry [1].

The semi-structured data documented in pathology reports contain crucial information regarding the biological and clinical features, histologic type, and TNM stage of cancers, which are essential for making optimal treatment decision making. These reports provide valuable information on the clinical and pathological features of the patient. To access these data in pathology reports, structured data required for the research of clinical oncology and cancer are contained in semi-structured text.

Traditionally, parsing pathology reports has been approached using regular expressions. However, these methods require labor-intensive and input from clinical experts or pathologists, and it becomes increasingly ineffective as the complexity between patterns of reports context increases dramatically as the counts of rule, and as underlying documents shift [2–5]. To overcome these limitations, artificial intelligence (AI) has been utilized to extract information from pathology reports [6,7]. Specifically, natural language processing (NLP) techniques, such as deep learning algorithms, have shown promise in this domain.

Although previous studies have structured pathology reports using NLP, the number of variables that can be extracted is limited [8–10]. For example, studies that extracted three to four variables, such as pathology grade, lesion location, and treatment method from pathology records or extract only biomarker information through the lymphoma classification tool, or immunopathology information, for a specific purpose have been mainly conducted [8].

One NLP algorithm that has gained significant attention is the bidirectional encoder representations from transformers (BERT) model, which has proven to be highly effective in various NLP models [11]. NLP has been applied for the extraction of pathology report information. In addition, many other parsing methods and tasks, such as a pre-trained model, have been applied to the analysis of pathology reports. For instance, BioBERT is a domain-specific language representation model pre-trained on a large-scale biomedical corpus [12]. Clinical-BERT is an accurate language model that captures physician-assessed semantic relationships in clinical texts [13]. Although previous studies have been conducted to obtain information of pathology reports using NLP, the variables can be limited.

In this context, we aimed to elucidate the deployment of deep learning data extraction methods for pathology reports in a single institute by addressing two issues. Our first objective

was to compare the performance of different NLP models, including BERT-basic, BioB-ERT, and ClinicalBERT, in terms of their accuracy in extracting information from pathology reports. Second objective was to compare the accuracy of regular expression method and NLP-based approaches for extracting information from pathology reports.

To achieve these objectives, we conducted a study using a dataset of pathology reports from a single institute. We compared the performance of different NLP models, including BERT-basic, BioBERT, and ClinicalBERT, in terms of their accuracy in extracting information from pathology reports. Additionally, we compared the accuracy of regular expression methods and NLP-based approaches, including BERT and its domain-specific variants, for extracting information from the dataset.

## Methods

### Data source

All electronically available pathology reports from the National Cancer Center were stored as a table in the Clinical Research Data Warehouse (CRDW). The CRDW database consists of medical records of patients with cancer from Electronic Health Records (EHR). The CRDW assigns the same anonymous identification key to each patient across all tables using a de-identification system [14]. Fig 1 showed that all electronically available pathology reports from the Breast Surgery Department of the National Cancer Center from January 1, 2005, to December 31, 2020, were retrieved. Our study encompassed a comprehensive dataset of 13,751 breast cancer surgical pathology reports. To ensure a manageable and representative sample for annotation, we employed a random selection process to extract approximately 10% of the reports (1,215 in total). Subsequently, we implemented a strategic division of these annotated reports, allocating them to training and test sets in a 7:3 ratio. This methodical approach resulted in a training set comprising 851 reports, while the remaining 364 reports were designated as the test set. This carefully curated dataset served as the foundation for developing and rigorously evaluating our pathology report extraction algorithm. The protocol was approved by the Institutional Review Board at National Cancer Center in Korea in April 2022 (NCC2022-0105). In the pathology reports, data have been accessed since 11/8/2022.

### Data labeling

To develop question-and-answer algorithms, the reference standards (i.e., an annotated corpus) needed to be obtained through breast cancer surgical pathology report review and in consultation with pathologists. Annotation guidelines were developed through repeated discussions with the pathologists to obtain standard annotations (S1 File). Following the annotation guideline, annotator worked with pathology reports in a structured interface divided into four main sections: documents list, annotation interface, setting screen, and annotation list (S1 Fig). Fig 2 shows that a health information manager with experience in pathology department work manually annotated 1,215 semi-structured pathology reports, using an open-source-based label studio. There was no separate preprocessing for negative terms due to the semi-structured nature of the data. Another individual cross-verified these annotations for accuracy. We annotated it using one annotator to maintain the consistency of the annotation. We focused on 19 phenotypes that explain the characteristics of the breast cancer surgical pathology report, including (1) organ, (2) tumor site, (3) histologic type, (4) intraductal component status, (5) nuclear grade, (6) necrosis status, (7) skin (nipple) invasion status, (8) lymph nodes, (9) anteriovenous invasion, (10) lymphovascular invasion, (11) tumor border, (12) microcalcification, (13) pathologic stage, (14) superior margin, (15) inferior margin, (16) medial margin, (17) lateral margin, (18) deep margin, and (19) superficial margin.

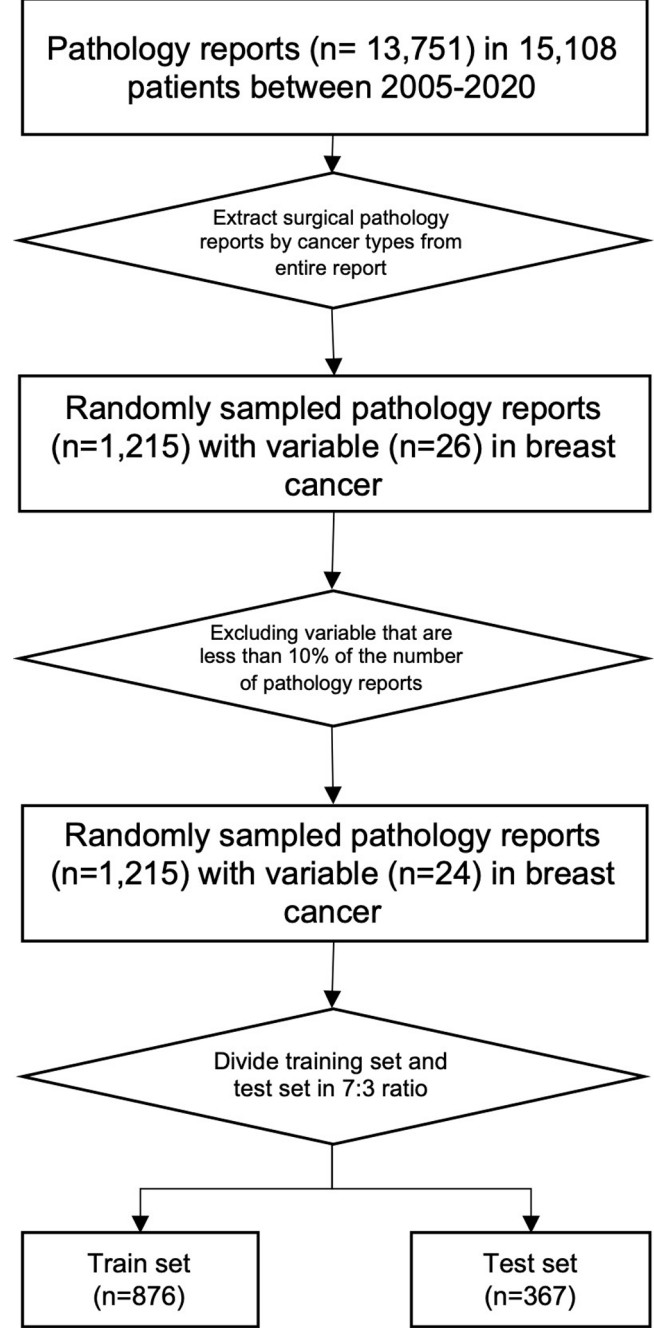

**Fig 1. Data flow chart of the study.**

## Data modeling

The BERT model was initially trained on the Books Corpus and Wikipedia [11]. However, it has limitations when it comes to learning professional terms, such as medical jargon. The language used in books and encyclopedias is more common, and technical terms such as medical jargon can be difficult to understand without professional education. To address this, ClinicalBERT and BioBERT were pretrained on technical terms, including medical jargon.

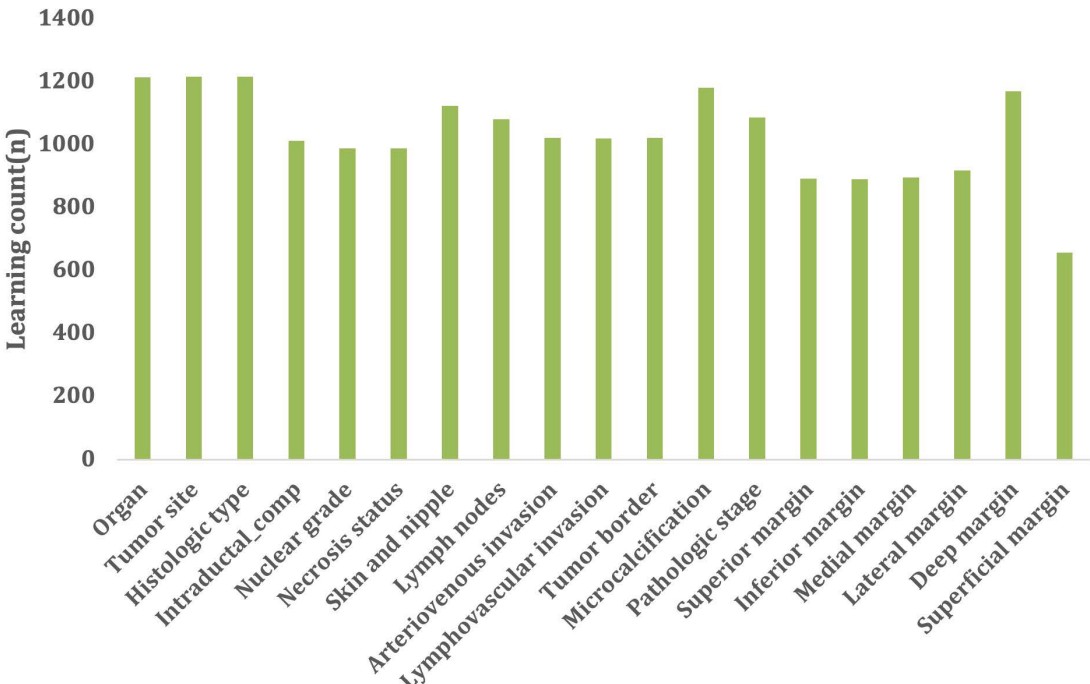

**Fig 2. Learning count by variable.**

BioBERT, for instance, uses transfer learning on BERT and was trained on nearly 18 billion words extracted from PubMed abstracts [12]. Similarly, ClinicalBERT was trained on BERT to improve its comprehension of clinical language [13].

BERT-based models can be fine-tuned based on annotation data to solve specific downstream tasks. We implemented a question-answering (QA) framework to extract information from pathology reports. This framework converts pathology report fields into questions and their corresponding values into answers, creating a structured approach to information extraction. For instance, when analyzing a pathology report, a field like "Tumor Site" becomes a question that seeks specific location information within the report. Our framework processes various types of clinical information, including tumor characteristics, histological details, and lymph node status. By using the SQuAD (Stanford Question Answering Dataset) architecture adapted for medical contexts, we developed a system that could understand and process medical terminology effectively [15]. Additionally, we followed the same pre-training process as established methods [16], which uses SQuAD. This approach allows the model to analyze pathology reports in a way that mirrors how medical professionals read and interpret these documents.

To verify the performance of BioBERT and ClinicalBERT, which were pretrained on the BERT model, K-fold cross-validation was used. This method divides the dataset into several groups and uses a test set by selecting one from each group. In addition, the result value obtained by repeating this process several times was calculated as an average and used as a verification result value. It is used to check whether the size of the collected data is small or if overfitting occurs in the learning process of the model.

For the major hyperparameters, the maximum sequence length was set to 128, the training batch size was set to 8, and the training epoch was set to 5. The hyperparameters were selected based on the computing power of the GPU resources. This experiment was performed in Python on 36 CPU cores, which are Intel ® Xeon ® Gold 6242R @ 3.1 GHz, 128 GB RAM,

and T100. We listed the average running times for each epoch of BERT, BioBERT, and ClinicalBERT.

The performance metrics for the NER (Named Entity Recognition) models were calculated using standard evaluation measures. In short, an extraction is considered a true positive when it successfully captures all essential information from the annotated text, even if the exact boundaries show minor variations. False positives occur when the extracted text either includes irrelevant information or fails to capture critical components of the annotation. Cases where the model fails to extract information or produces incomplete extractions are classified as false negatives. Precision, recall, and F1-score were computed for each entity type extracted from the pathology reports. Precision measures the proportion of correctly identified entities among all extracted entities, while recall measures the proportion of correctly identified entities among all actual entities in the dataset. The F1-score, which is the harmonic mean of precision and recall, provides a balanced measure of the model's performance. These metrics were calculated for BioBERT model and regular expression to assess their effectiveness in extracting clinical information from the pathology reports in Fig 3.

## Results

The optimization process of the model during the training procedure was investigated. The results, as depicted in Fig 4, indicate that both the output 1 loss and the output 2 loss rapidly decreased until the 5th epoch. The output 1 loss refers to the loss value of the start token, while the output 2 loss represents the loss value of the last token. The output loss, on the other hand, represents the total loss value, which was calculated by accumulating the cross entropy in the batch training process. Meanwhile, the output losses of 1 and 2 were calculated after completing the training. Both losses rapidly decreased until the 5th epoch.

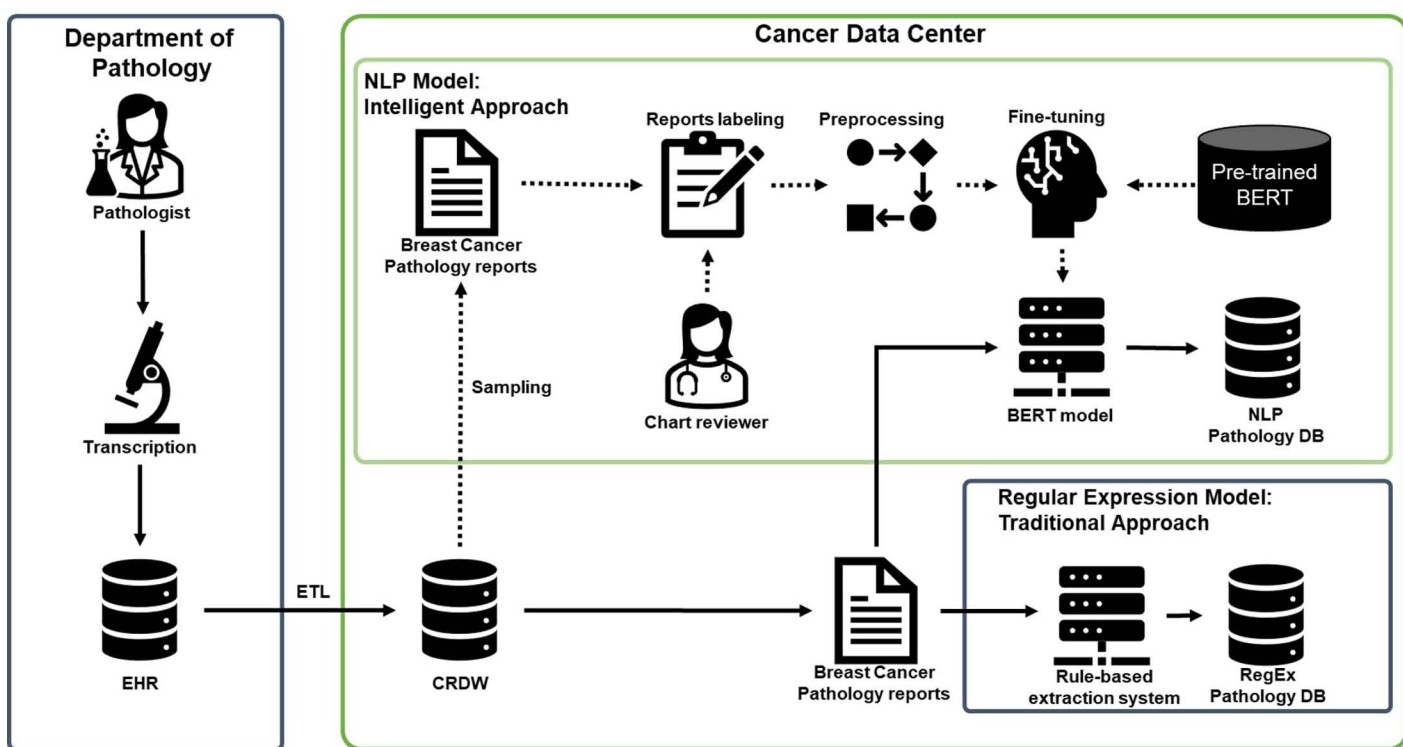

**Fig 3. BERT for clinical information from pathology report.**

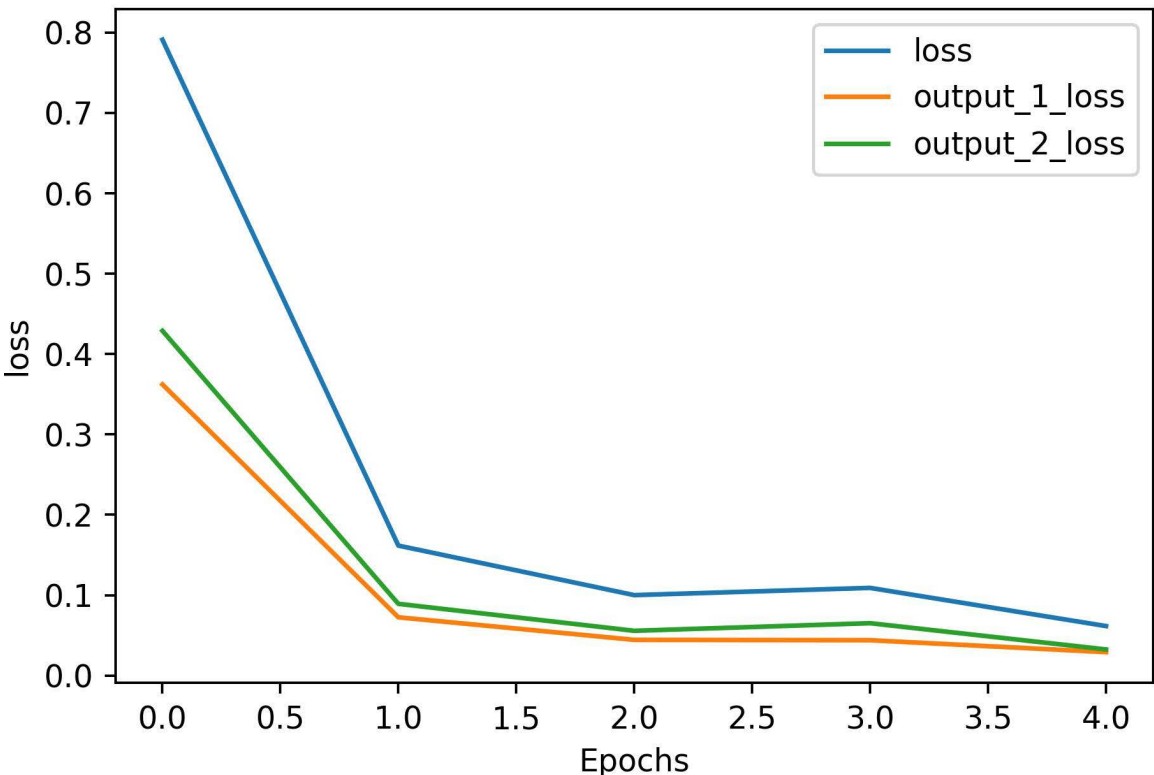

**Fig 4. Loss on the training and test sets according to the training set.**

Table 1 presents a comparative analysis of extraction performance for pathology reports across three models: BERT, BioBERT, and ClinicalBERT models. The results show the performance metrics obtained through 5-fold cross-validation, including both accuracy and F1-scores for each fold and their respective averages.

Overall, the results indicate that all the models demonstrated high accuracy in extracting information from pathology reports. However, BioBERT outperformed the other models, achieving the highest accuracy for all variables.

It was confirmed that the data were extracted with a high accuracy of 0.99 or more for most variables, but the deep margin (0.9631), lateral margin (0.9701), medial margin (0.9573), and superficial margin (0.9649) showed relatively low accuracy, as shown in Table 2. In the case of regular expressions, the margin could not be subdivided, so the accuracy could not be compared

**Table 1. Comparison of model performance metrics using 5-Fold cross-validation across different BERT variants.**

|  | BERT-BASE | | BioBERT | | ClinicalBERT | |
| --- | --- | --- | --- | --- | --- | --- |
|  | Accuracy | F1-score | Accuracy | F1-score | Accuracy | F1-score |
| K = 1 | 0.96955 | 0.97289 | 0.98515 | 0.98594 | 0.98908 | 0.99075 |
| K = 2 | 0.93369 | 0.92603 | 0.99029 | 0.99067 | 0.98411 | 0.98462 |
| K = 3 | 0.97837 | 0.97770 | 0.98514 | 0.98441 | 0.96036 | 0.96386 |
| K = 4 | 0.98312 | 0.98456 | 0.98316 | 0.98294 | 0.97920 | 0.98160 |
| K = 5 | 0.97618 | 0.97787 | 0.98669 | 0.98631 | 0.98394 | 0.98376 |
| average | 0.96818 | 0.96781 | 0.98609 | 0.98605 | 0.97934 | 0.98092 |

**Table 2. Extraction accuracy for validation.**

| Variable | BioBERT | | | | RegEx | | | |
|---|---|---|---|---|---|---|---|---|
| | Accuracy | Precision | Recall | F1 | Accuracy | Precision | Recall | F1 |
| Organ | 1 | 1 | 1 | 1 | 0.9751 | 0.9970 | 0.9751 | 0.9858 |
| Tumor site | 0.9903 | 0.9903 | 0.9903 | 0.9903 | 0.9339 | 0.8910 | 0.9339 | 0.9119 |
| Intraductal component | 1 | 1 | 1 | 1 | 0.1644 | 0.1653 | 0.1644 | 0.1649 |
| Histologic type | 0.9613 | 0.9570 | 0.9613 | 0.9583 | 0.8407 | 0.7267 | 0.8407 | 0.7780 |
| Nuclear grade | 0.9914 | 0.9871 | 0.9914 | 0.9892 | 0.9901 | 0.9910 | 0.9901 | 0.9904 |
| Necrosis status | 0.9957 | 0.9957 | 0.9957 | 0.9957 | 0.9950 | 0.9970 | 0.9950 | 0.9960 |
| Skin and nipple | 0.9928 | 0.9917 | 0.9928 | 0.9918 | 0.9567 | 0.9576 | 0.9567 | 0.9571 |
| Lymph nodes | 0.9886 | 0.9886 | 0.9886 | 0.9886 | 0.4792 | 0.6170 | 0.4792 | 0.4822 |
| Arteriovenous invasion | 0.9959 | 1 | 0.9959 | 0.9979 | 0.9786 | 0.9840 | 0.9786 | 0.9805 |
| Lymphovascular invasion | 0.9918 | 1 | 0.9918 | 0.9957 | 0.3759 | 0.3682 | 0.3759 | 0.3674 |
| Tumor border | 0.9921 | 0.9882 | 0.9921 | 0.9895 | 0.9942 | 0.9971 | 0.9942 | 0.9956 |
| Microcalcification | 1 | 1 | 1 | 1 | 0.1381 | 0.1314 | 0.1381 | 0.1307 |
| Pathologic stage | 0.9855 | 0.9927 | 0.9855 | 0.9888 | 0.9945 | 0.9982 | 0.9945 | 0.9961 |
| Superior margin | 1 | 1 | 1 | 1 | – | – | – | – |
| Inferior margin | 0.9871 | 0.9885 | 0.9871 | 0.9869 | – | – | – | – |
| Medial margin | 0.9573 | 0.9851 | 0.9573 | 0.9680 | – | – | – | – |
| Lateral margin | 0.9701 | 0.9775 | 0.9701 | 0.9697 | – | – | – | – |
| Deep margin | 0.9631 | 0.9923 | 0.9631 | 0.9733 | – | – | – | – |
| Superficial margin | 0.9649 | 0.9859 | 0.9649 | 0.9738 | – | – | – | – |

with NLP. The regular expression also showed an accuracy of 0.9 or higher, but showed low accuracy for values defined in various patterns such as intraductal component (0.1644), lymph node (0.4792), lymphovascular invasion (0.3759), and microcalcification (0.1381) variables.

As shown in Table 3, we also compared the results of extracting data using regular expressions and NLP from the same pathology report. Regular expression is limited to specific words or rules and provides limited information, such as 'Right', 'absent', and 'present'. On the other hand, NLP was able to obtain comprehensive information, such as 'right 3 o'clock chest wall' and 'no involvement of tumor'.

Table 3 shows that NLP performed better than regular expression in terms of the amount and quality of information extracted. For instance, NLP was able to extract more detailed information about the location of the tumor, the size of the tumor, and the type of carcinoma. In contrast, regular expression could only detect specific words or rules. Therefore, NLP has the potential to be a more powerful tool in data analysis and extraction.

## Discussion

We chose to implement NLP to enhance the existing standard method of data parsing, which is based on regular expressions. After conducting k-fold validation, we found that BioBERT is a highly accurate parsing method. Our findings confirmed that the NLP model-based parsing method yields data with higher overall accuracy compared to the existing method that uses regular expressions.

Several studies have explored NLP for electronic medical records, leveraging advanced models based on the existing BERT-based model. Pretrained medical terms such as Clinical-BERT and BioBERT have been developed. While BioBERT performed transfer learning on BERT using PubMed abstracts, ClinicalBERT tunes BERT for clinical language comprehension. Compared to BERT (accuracy: 0.96818, F1 score: 0.96781), it was confirmed that the

**Table 3. Regular expression and NLP result by variable.**

| Variable | Result of regular expression | Result of NLP |
|---|---|---|
| Location | Right | right 3 o'clock chest wall |
| | Right | right, inferior and lateral margins |
| | Left | left, lower outer |
| T-Size | 1.5 | 1.5 cm, 1.1 cm, & 0.7 cm |
| | 0.2 | 0.2 cm, residual |
| | 1.6 | up to 1.6 cm, multifocal |
| Skin and Nipple | absent | no involvement of tumor |
| | present | involvement of lactiferous duct |
| | present | Paget disease of nipple with involvement of lactiferous duct |
| Intraductal component | present | present, intratumoral (10%) |
| | present | present, intratumoral/extratumoral (40%) |
| | present | present, extratumoral (90%) |
| Lymphovascular Invasive | present | present, intratumoral/peritumoral |
| Histology type | Invasive lobular carcinoma | INVASIVE LOBULAR CARCINOMA, pleomorphic type associated with mixed lobular and ductal carcinoma in situ |
| | Mucinous carcinoma | MUCINOUS CARCINOMA associated with intracystic papillary carcinoma |
| Microcalcification | present | present, tumoral/non-tumoral |

accuracy of ClinicalBERT (accuracy: 0.97934, F1 score: 0.98092) and BioBERT (accuracy: 0.98609, F1 score: 0.98605) was relatively high, but BioBERT was considered the most suitable.

In our study, it was confirmed that BioBERT can obtain information from pathology reports with relatively higher accuracy than regular expression. In the case of lymph node, it had accuracy of 0.9886 in NLP, however showed a low accuracy of 0.4792 in regular expression. In the case of data in which the result value was not structurized such as lymph node and invasion information, it was considered that data could be collected with exceptionally high accuracy. Furthermore, our study demonstrated a significant advantage of the NLP model over regular expressions in extracting margin information. The complexity and variability of margin data in pathology reports posed a considerable challenge for rule-based approaches. While regular expressions struggled to capture this intricate information accurately, the NLP model, with its ability to understand context and nuance, successfully extracted and interpreted the margin data. This finding underscores the superiority of advanced NLP techniques in handling complex, unstructured medical text, particularly in cases where traditional methods fall short.

In contrast, regular expressions become more complex and diversified as data structuring rules become more elaborate, which leads to longer data pre-processing times [2,3,5]. Additionally, a regular expression is subject to the data developer's subjectivity in the design process, making it difficult to maintain in the future. While a regular expression parses data according to a set rule, it may lose information out of the rule. However, in the case of a learning model, data can be formalized by minimizing information loss according to an annotation method.

Previous studies have extracted necessary variables from pathology records through NLP; however, in this study, the number of imported variables was limited. For example, studies that extracted three to four variables, such as pathology grade, lesion location, and treatment method from pathology records or extract only biomarker information [4,8,10,17–19] through the lymphoma classification tool [20], or immunopathology information [21], for a specific purpose have been mainly conducted. Another study was limited in that large

amounts of data should be trained with high-performance computing power to increase accuracy. However, in our study, we were able to obtain a high-performance NLP model with a small amount of data using a model trained as the location value of the variable that needs to be extracted by selecting the pathology record paper.

Our study had several limitations. First, the model was developed using annotated pathology reports of breast cancer in a single institution, which might limit the generalizability of its application to other institutions. Second, because our study was conducted using only breast cancer pathology records, high accuracy cannot be expected when parsing the pathology record sheet of other cancer types with the relevant model.

This statement suggests that while the current study provides valuable insights into the potential of NLP for analyzing breast cancer data, there are still limitations to be addressed. To expand the scope of capabilities NLP in the medical field, future studies should consider applying this technology to a wider range of organ pathology records. By doing so, researchers will be able to identify patterns and relationships between different types of cancer and specific organs, leading to more accurate diagnoses and more targeted treatment plans.

Furthermore, it is worth noting that the NLP model used in this pilot study of breast cancer data offers a promising starting point for future research. However, it is important to acknowledge that there may be variations in the performance of NLP models across different types of data and medical contexts. As such, it may be necessary to further refine and adapt the NLP model to suit the unique needs of multi-organ pathology records. In addition to our ongoing work, we're looking into more in-depth analysis of advanced NLP models like pubmedBERT [22] and LLaMa [23]. This is to better address the specific needs of multi-organ pathology records.

In this study, we selected NLP to improve regular expression-based data structurization, which is an existing data parsing methodology. Our results demonstrate that BioBERT has high accuracy in pathology reports, and that the NLP model can obtain data with a higher overall accuracy than regular expressions. In conclusion, our findings suggest that the process of obtaining information from pathology reports should include NLP using BioBERT.

## Supporting information

**S1 File. Comprehensive annotation guideline for breast cancer surgical pathology reports.** (PDF)

**S1 Fig. Example of annotated pathology report for NLP task.** (PNG)

## Author contributions

**Conceptualization:** Phillip Park, Hyun-Jin Kim.

**Data curation:** Nayoung Han, Heejung Chae, Chong Woo Yoo.

**Formal analysis:** Yeonho Choi.

**Funding acquisition:** Hyun-Jin Kim.

**Investigation:** Juyeon Hwang.

**Methodology:** Yeonho Choi, Ye-Lin Park.

**Resources:** Kui Son Choi, Hyun-Jin Kim.

**Software:** Yeonho Choi.

**Supervision:** Kui Son Choi, Hyun-Jin Kim.

**Validation:** Nayoung Han, Heejung Chae, Chong Woo Yoo.

**Visualization:** Yeonho Choi.

**Writing – original draft:** Phillip Park.

**Writing – review & editing:** Phillip Park.

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
