## [Decision Letter · Decision Letter 0]

10 Sep 2024

PONE-D-24-22004Leveraging Natural Language Processing for Efficient Information Extraction from Breast Cancer Pathology Reports: Single-Institution StudyPLOS ONE

Dear Dr. Kim,

Thank you for submitting your manuscript to PLOS ONE. After careful consideration, we feel that it has merit but does not fully meet PLOS ONE’s publication criteria as it currently stands. Therefore, we invite you to submit a revised version of the manuscript that addresses the points raised during the review process.

Please find the reviewers comments below, I recommend you pay close attention to the concerns of reviewer 2. In your response to reviewers please comment on each point made by both reviewers. 

We look forward to receiving your revised manuscript.

Kind regards,

Joanna Tindall, PhD

Staff Editor

PLOS ONE

“This study was supported by a grant from the National Cancer Center (grant no. 1810871-2 and 2310400-1).”

4. In the online submission form you indicate that your data is not available for proprietary reasons and have provided a contact point for accessing this data. Please note that your current contact point is a co-author on this manuscript. According to our Data Policy, the contact point must not be an author on the manuscript and must be an institutional contact, ideally not an individual. Please revise your data statement to a non-author institutional point of contact, such as a data access or ethics committee, and send this to us via return email. Please also include contact information for the third party organization, and please include the full citation of where the data can be found.

Reviewers' comments:

Reviewer's Responses to Questions

**Comments to the Author**

1. Is the manuscript technically sound, and do the data support the conclusions?

Reviewer #1: Yes

Reviewer #2: No

2. Has the statistical analysis been performed appropriately and rigorously? 

Reviewer #1: Yes

Reviewer #2: N/A

3. Have the authors made all data underlying the findings in their manuscript fully available?

Reviewer #1: Yes

Reviewer #2: No

4. Is the manuscript presented in an intelligible fashion and written in standard English?

Reviewer #1: Yes

Reviewer #2: No

5. Review Comments to the Author

Reviewer #1: The study demonstrated that BioBERT, among the three BERT models evaluated, achieved the highest accuracy for extracting information from breast cancer pathology reports, with an average performance of 0.99901. However, there is some room for improvement.

*The paper states, "A total of 13,751 breast cancer surgical pathology reports were collected. Among these, 1,215 pathology reports were used to build annotated data to develop an extraction algorithm for pathology reports," but does not provide a reason why only 1,215 reports were used out of the 13,751 collected

*The paper provides an insufficient background and literature review. A more thorough review of existing methods and their limitations would strengthen the case for using BioBERT and provide a clearer context for the study.

*Figures such as the flowchart and Figure 3 are hard to understand. The flow and content of these figures are not clearly explained, making it difficult for readers to follow the methodology and results.

Reviewer #2: General comments

In this paper, the authors developed an information extraction model to extract key information from pathology reports. The authors compared the deep learning models with the regular expression method. I have some comments for this study.

Major comments

1. The innovation of methodology seems weak. What are the differences between this study and others?

2. The annotation guideline should be posted in the supplement.

3. Adding some annotation examples is recommended.

4. The authors should provide a detailed introduction about how they conduct the question-and-answer task when training IE models.

5. The cross-validation, hyperparameters, and hardware should be placed in an experimental setup section.

6. Why did the authors use the fuzzy-wuzzy algorithm to compare the performance? Why did the authors not compare the extracted results of these two methods directly?

7. The comparison between biobert and regular expression seems inappropriate.

8. As an IE task, precision, recall, F1 score should be reported. Using accuracy only is not enough.

6. PLOS authors have the option to publish the peer review history of their article (what does this mean? ). If published, this will include your full peer review and any attached files.

**Do you want your identity to be public for this peer review?** For information about this choice, including consent withdrawal, please see our Privacy Policy .

Reviewer #1: No

Reviewer #2: No

---

## [Author Response · Author response to Decision Letter 0]

5 Nov 2024

Response to Reviewer #1

Reviewer 1: The study demonstrated that BioBERT, among the three BERT models evaluated, achieved the highest accuracy for extracting information from breast cancer pathology reports, with an average performance of 0.99901. However, there is some room for improvement.

We appreciate the comprehensive review. We have attempted to revise our manuscript in accordance with the reviewer’s suggestions and comments. Our responses to each of the points raised are provided below.

Comments:

Comment 1. The paper states, "A total of 13,751 breast cancer surgical pathology reports were collected. Among these, 1,215 pathology reports were used to build annotated data to develop an extraction algorithm for pathology reports," but does not provide a reason why only 1,215 reports were used out of the 13,751 collected.

Thank you for your detailed comment. As commented by you, we have provided a reason why 1,215 reports were used out of the 13,751 collected in the Methods section of the revised manuscript as follows:

“Our study encompassed a comprehensive dataset of 13,751 breast cancer surgical pathology reports. To ensure a manageable and representative sample for annotation, we employed a random selection process to extract approximately 10% of the reports (1,215 reports in total). Subsequently, we implemented a strategic division of these annotated reports, allocating them to training and test sets in a 7:3 ratio. This methodical approach resulted in a training set comprising 851 reports, while the remaining 364 reports were designated as the test set. This carefully curated dataset served as the foundation for developing and rigorously evaluating our pathology report extraction algorithm.”

Comment 2. The paper provides an insufficient background and literature review. A more thorough review of existing methods and their limitations would strengthen the case for using BioBERT and provide a clearer context for the study.

In response to your comment, we have described the details about the BERT models including BioBERT and ClinicalBERT in the Background section as follows:

“Our primary object was to evaluate the efficacy of various NLP models, specifically BERT-basic, BioBERT, and ClinicalBERT, in accurately extracting information from pathology reports. The selection of these BERT-based models was informed by their specialized pre-training: BioBERT, having been pre-trained on biomedical corpora, is particularly adept at biomedical text analysis, while ClinicalBERT, pre-trained on clinical texts, potentially offers the enhanced performance on clinical documents such as pathology reports. For example, BioBERT has demonstrated exceptional capabilities in biomedical named entity recognition tasks, whereas ClinicalBERT has shown proficiency in extracting clinical concepts.”

Comment 3. Figures such as the flowchart and Figure 3 are hard to understand. The flow and content of these figures are not clearly explained, making it difficult for readers to follow the methodology and results.

Thank you for your appropriate comment. According to your suggestions, we have clearly revised the contents of Figures. In addition, more detailed explanation of the figure's content and flow have been added in Method section. We hope that these changes will help readers better understand the methodology and results of this study.

Please refer to the revised Figures.

Your feedback has been instrumental in enhancing the overall quality and accessibility of our paper, and we are grateful for your constructive criticism. 

Response to Reviewer #2

Reviewer 2: In this paper, the authors developed an information extraction model to extract key information from pathology reports. The authors compared the deep learning models with the regular expression method. I have some comments for this study.

We appreciate the comprehensive review. We have attempted to revise our manuscript in accordance with the reviewer’s suggestions and comments. Our responses to each of the points raised are provided below.

Comments:

Comment 1. The innovation of methodology seems weak. What are the differences between this study and others?

Thank you for your appropriate comment. We agree with your comment that the innovation of methodology seems weak. However, the purpose of this study was not to develop a novel NLP method, but rather to evaluate and select the most optimal model for our specific pathology reports. We utilized existing methodologies, particularly BERT-based models, focusing on adapting and optimizing them for our unique dataset. This approach allowed us to leverage proven techniques while tailoring them to our specific needs.

By comparing multiple BERT-based models (BioBERT, ClinicalBERT, and BERT-basic), we were able to identify the most effective model for our particular use case. This comparative analysis and optimization process is a key differentiator of our study, as it provides valuable insights into the performance of these models in the context of breast cancer pathology reports.

In addition, there are differences in the results between this study and other studies. Previous studies using pre-BERT models mainly focused on extracting information from a limited set of fields in pathology reports, whereas our study demonstrates a key strength in its ability to extract information from a wide range of fields. This comprehensive approach allows for a more accurate analysis of clinicopathological parameters in breast cancer patients.

We hope that this will be accepted.

Comment 2. The annotation guideline should be posted in the supplement.

It is a good comment which needs to be considered. As commented, we have added the annotation guideline (S1 File. Comprehensive annotation guideline for breast cancer surgical pathology reports) and this content has been also described in Methods section.

Please refer to the S1 file.

Comment 3. Adding some annotation examples is recommended.

We appreciate your suggestion to include annotation examples. In response to your recommendation, we have created annotation examples and added them as a supporting information figure (S1 Fig) to our paper. These examples will help readers understand our annotation process more clearly.

Please refer to the S1 Fig.

Comment 4. The authors should provide a detailed introduction about how they conduct the question-and-answer task when training IE models.

According to your suggestion, the following description has been added in the second paragraph of the Data modeling.

“BERT-based models can be fine-tuned based on annotation data to solve specific downstream tasks. In this study, the task of extracting unstructured information was framed as a question-and-answer format, identifying 19 types of entities. To fine-tune for Question and Answer, we used the same BERT architecture used for SQuAD [12], Additionally, we followed the same pre-training process as established methods [13], which uses SQuAD.”

Comment 5. The cross-validation, hyperparameters, and hardware should be placed in an experimental setup section.

Thank you for your feedback. As commented, the information about cross-validation, hyperparameters, and hardware specifications have been moved to the methods section, specifically under a data modeling

Comment 6. Why did the authors use the fuzzy-wuzzy algorithm to compare the performance? Why did the authors not compare the extracted results of these two methods directly?

Thank you for your appropriate comment regarding our performance evaluation methodology. We would like to clarify that we did not use the fuzzy-wuzzy algorithm for comparing performance. To avoid confusion for readers, we have deleted the fuzzy-wuzzy algorithm and have clarified Named Entity Recognition (NER) for our performance evaluation in Methods and Results sections as follows:

“The performance metrics for the NER (Named Entity Recognition) models were calculated using standard evaluation measures. Precision, recall, and F1-score were computed for each entity type extracted from the pathology reports. Precision measures the proportion of correctly identified entities among all extracted entities, while recall measures the proportion of correctly identified entities among all actual entities in the dataset. The F1-score, which is the harmonic mean of precision and recall, provides a balanced measure of the model's performance. These metrics were calculated for BioBERT model and regular expression to assess their effectiveness in extracting clinical information from the pathology reports in Fig 3.”

Comment 7. The comparison between biobert and regular expression seems inappropriate.

Thank you for your detailed comment. We agree with your comment that the direct comparison between BioBERT and regular expression seems inappropriate. However, in some respects, the comparison between the two methods may be worthwhile for the following reasons:

To extract information from pathology reports, we initially used the regular expression approach due to its simplicity and low implementation barrier (revised Figure 3). However, as data complexity increases, rule-based systems often required the extensive exception handling, leading to decreased accuracy and increased maintenance challenges. For these reasons, this study aimed to establish a natural language processing model approach that overcomes the limitations of the existing traditional model, regular expressions, by comparing the natural language processing model with existing processed regular expressions.

Our findings may provide valuable insights into the importance of an advanced NLP models approach, compared to traditional regular expressions when extracting information from cancer pathology reports.

We hope that this will be accepted.

Comment 8. As an IE task, precision, recall, F1 score should be reported. Using accuracy only is not enough.

Thank you for your valuable feedback regarding the performance metrics. As commented, we have reported not only accuracy but also precision, recall, and F1 score in our performance evaluation.

Please refer to the revised Table 2.

---

## [Decision Letter · Decision Letter 1]

5 Dec 2024

PONE-D-24-22004R1Leveraging Natural Language Processing for Efficient Information Extraction from Breast Cancer Pathology Reports: Single-Institution StudyPLOS ONE

Dear Dr. Kim,

Thank you for submitting your manuscript to PLOS ONE. After careful consideration, we feel that it has merit but does not fully meet PLOS ONE’s publication criteria as it currently stands. Therefore, we invite you to submit a revised version of the manuscript that addresses the points raised during the review process.

We look forward to receiving your revised manuscript.

Kind regards,

Paolo Torroni

Academic Editor

PLOS ONE

Additional Editor Comments:

While the authors did reply to reviewer comments in their rebuttal, the way that reflects in the final manuscript is not yet acceptable. The paper does not yet make a convincing case for its significance with respect to the state of the art, misses important details, and does not properly include exemplifications as requested. Please include all requested details in the paper, to make the contribution self-contained and the work reproducible, with a clear description of the experimental methodology and of the evaluation criteria.

Reviewers' comments:

Reviewer's Responses to Questions

**Comments to the Author**

1. If the authors have adequately addressed your comments raised in a previous round of review and you feel that this manuscript is now acceptable for publication, you may indicate that here to bypass the “Comments to the Author” section, enter your conflict of interest statement in the “Confidential to Editor” section, and submit your "Accept" recommendation.

Reviewer #2: All comments have been addressed

Reviewer #3: (No Response)

2. Is the manuscript technically sound, and do the data support the conclusions?

Reviewer #2: Yes

Reviewer #3: Partly

3. Has the statistical analysis been performed appropriately and rigorously? 

Reviewer #2: N/A

Reviewer #3: N/A

4. Have the authors made all data underlying the findings in their manuscript fully available?

Reviewer #2: No

Reviewer #3: No

5. Is the manuscript presented in an intelligible fashion and written in standard English?

Reviewer #2: Yes

Reviewer #3: Yes

6. Review Comments to the Author

Reviewer #2: Thanks for considering my comments. All my concerns have been addressed. I think this paper is ready for acceptance.

Reviewer #3: The authors addressed most of the reviewers' comments and questions. However, some important information is still missing.

- Reviewer 1, comment 2 ("The paper provides an insufficient background and literature review."): A "Related work" section is still missing. The authors should cite some similar works on information extraction in the biomedical field and highlight the strengths of their work in comparison.

- Reviewer 2, comment 3: the authors added an annotation example as a supplementary file. They should instead include it in the paper and appropriately comment and explain it.

- Reviewer 2, comment 4: the authors added some information on how they conducted the question-and-answer task when training IE models. However, they do so by citing another paper, while they should also add an explanation for those who aren't familiar with that paper.

- They should also explain how they evaluate models’ answers. In an IE extraction task, it is important to explain what is considered a false positive/negative if/when the extracted text partially match the annotations.

- More generally, it is still unclear how the task is really conducted. The authors should include some examples of questions/answers used to train the models and explain what is given as input to the models during evaluation.

- Were Bert models fine-tuned without a validation set? Only train/test? Why?

- What is the metric reported in Table 1? Accuracy or F1? In the data modelling paragraph, the authors talk about F1, while in the Result paragraph they talk about accuracy and the table doesn't specify it.

Minor issues:

The formatting doesn't look right to me, did you use the official latex template? Also, images are really low quality and should be included in a higher resolution.

Revise grammar, expecially in the Introduction:

- Traditionally, parsing pathology reports has been approached using regular EXPRESSIONS

- However, THESE methods REQUIRE

- which has proven to be highly effective in various NLP MODELS

- Although PREVIOUS studies

The last two paragraphs of the introduction are almost exact replicas.

7. PLOS authors have the option to publish the peer review history of their article (what does this mean? ). If published, this will include your full peer review and any attached files.

**Do you want your identity to be public for this peer review?** For information about this choice, including consent withdrawal, please see our Privacy Policy .

Reviewer #2: No

Reviewer #3: No

---

## [Author Response · Author response to Decision Letter 1]

5 Jan 2025

[Responses to comments]

Response to Reviewer #2

Reviewer 2: Thanks for considering my comments. All my concerns have been addressed. I think this paper is ready for acceptance.

Thank you very much for your thorough review and positive feedback. We greatly appreciate your time and effort in evaluating our manuscript. Your constructive comments have helped us improve the quality of our paper significantly. We are grateful for your recommendation for acceptance. 

Response to Reviewer #3

Reviewer 3: The authors addressed most of the reviewers' comments and questions. However, some important information is still missing.

We appreciate the comprehensive review. We have attempted to revise our manuscript in accordance with the reviewer’s suggestions and comments. Our responses to each of the points raised are provided below.

Comments:

Comment 1. Reviewer 1, comment 2 ("The paper provides an insufficient background and literature review."): A "Related work" section is still missing. The authors should cite some similar works on information extraction in the biomedical field and highlight the strengths of their work in comparison.

In response to your comment, we have described the details about previous studies in the Background section as follows:

“Although previous studies have structured pathology reports using NLP, the number of variables that can be extracted is limited.8-10 For example, studies that extracted three to four variables, such as pathology grade, lesion location, and treatment method from pathology records or extract only biomarker information through the lymphoma classification tool, or immunopathology information, for a specific purpose have been mainly conducted.8”

8. Kim Y, Lee JH, Choi S, et al. Validation of deep learning natural language processing algorithm for keyword extraction from pathology reports in electronic health records. Scientific reports. 2020;10(1):20265.

9. Kefeli J, Berkowitz J, Acitores Cortina JM, Tsang KK, Tatonetti NP. Generalizable and automated classification of TNM stage from pathology reports with external validation. Nature Communications. 2024;15(1):8916.

10. Buckley JM, Coopey SB, Sharko J, et al. The feasibility of using natural language processing to extract clinical information from breast pathology reports. Journal of pathology informatics. 2012;3(1):23.

Comment 2. The authors added an annotation example as a supplementary file. They should instead include it in the paper and appropriately comment and explain it.

It is a good comment which needs to be considered. As commented, we have explained the annotation guideline (S1 File. Comprehensive annotation guideline for breast cancer surgical pathology reports) and example of annotation (S1 Fig. Example of annotated pathology reports for NLP task) this content has been also described in Methods section.

“Annotation guidelines were developed through repeated discussions with the pathologists to obtain standard annotations (S1 File). Following the annotation guideline, annotator worked with pathology reports in a structured interface divided into four main sections: documents list, annotation interface, setting screen, and annotation list (S1 Fig).”

Comment 3. The authors added some information on how they conducted the question-and-answer task when training IE models. However, they do so by citing another paper, while they should also add an explanation for those who aren't familiar with that paper.

According to your suggestion, the following description has been added in the second paragraph of the Data modeling.

“We implemented a question-answering (QA) framework to extract information from pathology reports. This framework converts pathology report fields into questions and their corresponding values into answers, creating a structured approach to information extraction. For instance, when analyzing a pathology report, a field like "Tumor Site" becomes a question that seeks specific location information within the report. Our framework processes various types of clinical information, including tumor characteristics, histological details, and lymph node status. By using the SQuAD (Stanford Question Answering Dataset) architecture adapted for medical contexts, we developed a system that could understand and process medical terminology effectively [15]. Additionally, we followed the same pre-training process as established methods [16], which uses SQuAD. This approach allows the model to analyze pathology reports in a way that mirrors how medical professionals read and interpret these documents.”

Comment 4. They should also explain how they evaluate models’ answers. In an IE extraction task, it is important to explain what is considered a false positive/negative if/when the extracted text partially match the annotations.

Thank you for your appropriate comment regarding our performance evaluation methodology. we have clarified Named Entity Recognition (NER) for our performance evaluation in Methods sections as follows:

“The performance metrics for the NER (Named Entity Recognition) models were calculated using standard evaluation measures. In short, an extraction wass considered a true positive when it successfully captures all essential information from the annotated text, even if the exact boundaries show minor variations. False positives occur when the extracted text either includes irrelevant information or fails to capture critical components of the annotation. Cases where the model fails to extract information or produces incomplete extractions are classified as false negatives.”

Comment 5. More generally, it is still unclear how the task is really conducted. The authors should include some examples of questions/answers used to train the models and explain what is given as input to the models during evaluation.

Thank you for your feedback. As commented, the example about question/answers framework have been added to the methods section as follows:

“For instance, when analyzing a pathology report, a field like "Tumor Site" becomes a question that seeks specific location information within the report.”

Comment 6. Were Bert models fine-tuned without a validation set? Only train/test? Why?

Thank you for your question about our validation approach. We did not use a separate validation set, due to the small sample size. However, as an alternative, we employed k-fold cross-validation for model evaluation. Through k-fold cross-validation, we were able to effectively assess our model's performance using only train and test sets, as the cross-validation process itself provides robust validation of the model's performance across multiple data splits.

This approach allowed us to maximize the use of our available data while maintaining rigorous evaluation standards. The final performance metrics reported in our study represent the averaged results across all k-fold iterations, ensuring reliable and generalizable results.

Comment 7. What is the metric reported in Table 1? Accuracy or F1? In the data modelling paragraph, the authors talk about F1, while in the Result paragraph they talk about accuracy and the table doesn't specify it.

Thank you for bringing up this important point about the metrics in Table 1. We have addressed this by clearly specifying both accuracy and F1 scores in the table and ensuring consistency throughout the paper. In the data modeling section, we have discussed F1 scores as they provide a balanced measure of precision and recall. We have also present both accuracy and F1 scores to give a comprehensive view of our models' performance in the Results section. In addition, we have updated Table 1 to clearly label these metrics and maintain consistency with the discussion in both sections.

“Compared to BERT (accuracy: 0.96818, F1 score: 0.96781), it was confirmed that the accuracy of ClinicalBERT (accuracy: 0.97934, F1 score: 0.98092) and BioBERT (accuracy: 0.98609, F1 score: 0.98605) was relatively high, but BioBERT was considered the most suitable.”

---

## [Decision Letter · Decision Letter 2]

22 Jan 2025

Leveraging Natural Language Processing for Efficient Information Extraction from Breast Cancer Pathology Reports: Single-Institution Study

PONE-D-24-22004R2

Dear Dr. Kim,

We’re pleased to inform you that your manuscript has been judged scientifically suitable for publication and will be formally accepted for publication once it meets all outstanding technical requirements.

Kind regards,

Sergio Consoli

Academic Editor

PLOS ONE

Additional Editor Comments (optional):

Reviewers' comments:

Reviewer's Responses to Questions

**Comments to the Author**

1. If the authors have adequately addressed your comments raised in a previous round of review and you feel that this manuscript is now acceptable for publication, you may indicate that here to bypass the “Comments to the Author” section, enter your conflict of interest statement in the “Confidential to Editor” section, and submit your "Accept" recommendation.

Reviewer #2: All comments have been addressed

Reviewer #3: All comments have been addressed

2. Is the manuscript technically sound, and do the data support the conclusions?

Reviewer #2: Yes

Reviewer #3: (No Response)

3. Has the statistical analysis been performed appropriately and rigorously? 

Reviewer #2: Yes

Reviewer #3: (No Response)

4. Have the authors made all data underlying the findings in their manuscript fully available?

Reviewer #2: No

Reviewer #3: (No Response)

5. Is the manuscript presented in an intelligible fashion and written in standard English?

Reviewer #2: Yes

Reviewer #3: (No Response)

6. Review Comments to the Author

Reviewer #2: (No Response)

Reviewer #3: (No Response)

7. PLOS authors have the option to publish the peer review history of their article (what does this mean? ). If published, this will include your full peer review and any attached files.

**Do you want your identity to be public for this peer review?** For information about this choice, including consent withdrawal, please see our Privacy Policy .

Reviewer #2: No

Reviewer #3: No

---

## [Editor Report · Acceptance letter]

PONE-D-24-22004R2

PLOS ONE

Dear Dr. Kim,

I'm pleased to inform you that your manuscript has been deemed suitable for publication in PLOS ONE. Congratulations! Your manuscript is now being handed over to our production team.

Kind regards,

on behalf of

Dr. Sergio Consoli

Academic Editor

PLOS ONE